# The Relationship between COVID-19 Protection Behaviors and Pandemic-Related Knowledge, Perceptions, Worry Content, and Public Trust in a Turkish Sample

**DOI:** 10.3390/vaccines10122027

**Published:** 2022-11-27

**Authors:** Melike Kucukkarapinar, Filiz Karadag, Irem Budakoglu, Selcuk Aslan, Onder Ucar, Aysegul Yay Pence, Utku Timurcin, Selim Tumkaya, Cicek Hocaoglu, Ilknur Kiraz

**Affiliations:** 1Faculty of Medicine, Psychiatry Department, Gazi University, Ankara 06560, Turkey; 2Neuroscience and Neurotechnology Center of Excellence (NÖROM), Ankara 06560, Turkey; 3Faculty of Medicine, Department of Medical Education and Informatics, Gazi University, Ankara 06560, Turkey; 4Faculty of Medicine, Gazi University, Phase VI, Ankara 06560, Turkey; 5Faculty of Medicine, Psychiatry Department, Pamukkale University, Denizli 20070, Turkey; 6Faculty of Medicine, Psychiatry Department, Recep Tayyip Erdogan University, Rize 53020, Turkey

**Keywords:** COVID-19, perception, behavior, preventive measures, worry, trust, conspiracy theory

## Abstract

Background: This study aimed to explore the effect of knowledge, COVID-19-related perceptions, and public trust on protective behaviors in Turkish people. Methods: Data were collected from an online survey (Turkish COVID-19 Snapshot Monitoring) conducted between July 2020 and January 2021. The recommended protective behaviors (hand cleaning, wearing a face mask, and physical distancing) to prevent COVID-19 were examined. The impacts of the following variables on protective behaviors were investigated using logistic regression analysis: knowledge, cognitive and affective risk perception, pandemic-related worry content, public trust, conspiracy thinking, and COVID-19 vaccine willingness. Results: Out of a total of 4210 adult respondents, 13.8% reported nonadherence to protection behavior, and 86.2% reported full adherence. Males and young (aged 18–30 years) people tend to show less adherence. Perceived self-efficacy, susceptibility, and correct knowledge were positively related to more adherence to protective behavior. Perceptual and emotional factors explaining protective behavior were perceived proximity, stress level, and worrying about the relatives who depended on them. Trust in health professionals and vaccine willingness were positive predictors, while conspiracy thinking and acquiring less information (<2, daily) were negative predictors. Unexpectedly, trust in the Ministry of Health showed a weak but negative association with protection behavior. Conclusions: Perceived stress, altruistic worries, and public trust seem to shape protection behaviors in addition to individuals’ knowledge and cognitive risk perception in respondents. Males and young people may have a greater risk for nonadherence. Reliable, transparent, and culture-specific health communication that considers these issues is required.

## 1. Introduction

The COVID-19 pandemic began in early 2020 and has quickly become a major public-health issue. At the beginning, because of the absence of vaccines or effective treatment options, the WHO announced that protective measures were fundamental to control the disease [1]. These measures fell into two categories: one of them included handwashing, cleaning surfaces, wearing a face mask in public, not touching your eyes, mouth, or nose, and sneezing into tissues; and the second was to maintain physical distance from others [2]. Currently, the effectiveness of vaccines against new virus variants is uncertain, new treatment options are not widely available, and adequate vaccination is lacking in some countries.; therefore, hygiene practices and maintaining physical distance still maintain their importance.

Several studies on the current pandemic have revealed that two dimensions of risk perception (cognitive and affective) and public awareness are important to influence people to take precautions [3]. Their results suggest that cognitive risk perceptions (susceptibility, severity, and perceived self-efficacy) and preparedness may help predict adherence to preventive measures [4]. Several studies have also found that affective risk perceptions, such as pandemic-related negative emotions [5], anxiety [6,7], and worry [8], were found to predict protective behaviors. Concern for family members increases fear of COVID-19 because of its fatality and unpredictable nature [9]. Pandemic-induced stress may influence COVID-19 protective behavior by having an impact on the health of family members and other significant people, as well as community life changes, such as mandatory quarantines, social restrictions, and job loss.

The second important factor is public awareness of the pandemic, which influences actual behavior during the pandemic [10,11]. Obtaining accurate information and disseminating it through responsible institutions and media can help increase public awareness and behavioral adoption [11]. Depending on the public’s trust in institutions, the public may adopt institution-recommended preventive measures. Belief in institutions and decision-makers is critical in a pandemic because it affects how public messages are heard, interpreted, and responded to [12,13]. A lack of trust can skew information interpretation, raise concerns, and alter the course of action [14]. Vaccine hesitancy and low compliance with official guidelines have been linked to trusting conspiracy theories rather than reliable sources of information [15].

Finally, in psychological models of behavior change, sociodemographic factors, such as education, age, sex, and culture, were deemed less important than risk perception and knowledge [16]. During the COVID-19 pandemic, recent research revealed that male sex and younger age were associated with less adherence to protective behavior [17,18]. This issue could be crucial in Turkey, which has a sizable youth population.

To develop effective strategies to reduce infection spread, it is critical to assess which factors influence people’s behavior during a pandemic. During the COVID-19 pandemic, we wanted to see how risk perception, pandemic-related emotions, trust in institutions, conspiracy theories, and demographic factors influenced protective behaviors (hand washing, mask wearing, and physical distance). This study aimed to answer the following questions: Do protective behaviors vary by age, gender, and educational level? What is the relationship between risk perception, knowledge, and COVID-19 protection behavior? Are people’s COVID-19 protection behaviors influenced by their worry content related to the pandemic? Can trust in institutions and conspiracy theories affect COVID-19 protection behavior?

In addition, identifying preventive behavior determinants in Turkish adults may help develop measures or policies to increase the practice of this protective behavior. In the event of future pandemics, this may help provide healthier conditions.

## 2. Material and Methods

In this study, the COVID-19 pandemic questionnaire created by WHO researchers in line with the literature was used [19]. The items were used to measure respondents’ attitudes toward the COVID-19 pandemic [20].

### 2.1. Survey Design

This cross-sectional study was carried out using a snowball technique on adult (over 18 years of age) Turkish residents between July 2020 and January 2021.

The questionnaire invitation link (hosted by Google) was sent to respondents via official email addresses of academic institutions, municipal councils, organizations, and worksites, as well as social-networking sites. Participants’ permission was sought at the start of the survey. The questionnaires were filled out anonymously. Respondents who answered all of the survey were included in the study. The detailed study protocol was described in our previous study [20].

### 2.2. Measures

#### 2.2.1. Sociodemographic Characteristics

Age, sex, educational status, family size, presence of chronic disease (i.e., obesity, diabetes, hypertension, and chronic obstructive pulmonary disease), being infected with COVID-19, and knowing people infected with COVID-19 in their intimate social environment were all factors to consider [20].

#### 2.2.2. COVID-19 Protection Behaviors

The COVID-19 protection behavior outcome or dependent variable was to perform all these behaviors: handwashing (for 20 s), using a disinfectant when handwashing is not possible, avoiding touching the face, eyes, or nose without washing hands, using a face mask, and maintaining physical distance in social interactions or public places. Response options were dichotomic “yes” or “no”. These items were modified from an H1N1 pandemic study [21].

#### 2.2.3. Knowledge about COVID-19 Infection

The correct knowledge level is assessed using five items, two of which are multiple-choice questions: the incubation period of the COVID-19 virus and whether a treatment or vaccine is available for COVID-19 infection. The remaining items involve hand hygiene, the use of masks, and physical distance. The response “this has nothing to do with the protection against coronavirus infection” was deemed incorrect. The knowledge was evaluated in a binary manner. Only participants who answered correctly on all five items were considered to have “correct information”.

#### 2.2.4. Risk Perception and Self-Efficacy

Cognitive risk perception (severity, susceptibility, and self-efficacy) [22] was measured using validated study items on a seven-point Likert scale. The relevant questions are presented in Electronic Appendix A; for example, the following question was rated between ‘not severe’ and ‘very severe’ to assess perceived severity: “How severe would it be for you to contract the novel coronavirus?”

Affective risk perception was assessed using eight items (e.g., perceived proximity, fear, anxiety, stress, and feeling helpless). An example of these items was that “The novel coronavirus to me feels…” ranged from “very close to me” to “too far away from me” [23]. Worry domains, as another part of affective risk perception, were evaluated utilizing crisis-specific items adopted from the Worry Domains Questionnaire [24] (e.g., ‘At the moment, how much do you worry about the loss of loved ones?’, ranging from ‘do not worry at all’ to ‘worry a lot’). All measures were rated on a seven-point Likert-type scale. The relevant items are presented in detail in Appendix A.

#### 2.2.5. Trust in Institutions and Public Attitudes toward COVID-19 Vaccines

Public trust was assessed by questioning trust in institutions (media, Ministry of Health, hospitals, and Turkish Medical Association) to combat the current pandemic and trust in information provided by health-care professionals. The possible responses ranged from “I completely trust” to “I do not trust at all”. The attitudes toward the COVID-19 vaccines were investigated using the following question: “If a coronavirus vaccine is discovered and recommended for me, I will get the vaccine” on a seven-point Likert scale, with answers ranging from “I completely agree” to “I do not agree at all” [25].

#### 2.2.6. Beliefs in Conspiracy Theories

The Conspiracy Mentality Questionnaire [26], a seven-point Likert-type scale with five items, was used to assess general belief in conspiracy theories. The arithmetic mean of five items was calculated to produce a final scale score for each participant.

## 3. Statistical Analysis

SPSS software version 22 was used to analyze the data. The chi-square test and independent samples *t* tests were used to compare respondents in relation to the variables mentioned in the Methods section; additionally, to explore the predictors of COVID-19 protection behaviors, multivariate logistic regression was performed. Independent variables were chosen based on the results of univariate analysis. Considering their importance in the current pandemic, the level of trust in health professionals and health authorities was also included in the regression model. The odds ratios (ORs) and 95% confidence intervals (CIs) were calculated. A *p* value lower than 0.05 was used to determine statistical significance.

## 4. Results

### 4.1. Characteristics of the Sample

The sample included 4210 people (2532 females and 1678 males; M age = 35.86, SD age = 13.75). Table 1 shows the respondents’ sociodemographic characteristics.

### 4.2. Sociodemographic Characteristics and COVID-19 Protection Behaviors

In the overall sample, 13.8% (*n* = 579) of respondents reported nonadherence to COVID-19 protection behaviors, while 86.2% (*n* = 3631) of those reported full adherence. Males, younger age groups, and less educated respondents reported significantly less adherence to protection behavior than their counterparts (Table 2).

Knowing accurate information about COVID-19 and acquiring less information (<2 times daily) about COVID-19 have been associated with COVID-19 protection behaviors (Table 1).

### 4.3. Risk Perception and Self-Efficacy

The respondents who adhered to COVID-19 protection behaviors perceived more self-efficacy, had higher scores of the perceived susceptibility and perceived severity of COVID-19 infection, perceived themselves as closer to COVID-19, and perceived that the pandemic spread faster than the nonadherence group (Table 2).

The willingness to be vaccinated against COVID-19 and not believing the media’s exaggeration were significantly higher in the fully adherent group than in the nonadherent group (Table 2).

Those who adhered to the COVID-19 protection behaviors had strikingly higher mean scores of negative emotions (fear, anxiety, and stress); additionally, the mean scores of the pandemic-related worry domains were significantly higher in the first group than in the second group, especially worries about oneself or loved ones’ physical and mental health, overload of health systems, and access to food supplies. The levels of worry about financial issues were also significantly different between groups (Table 2).

### 4.4. Trust in Institutions and Conspiracy Theories Related to Pandemic

Respondents who engaged in COVID-19 protection behaviors trusted in the media, health institutions such as the Turkish Medical Association and hospitals, and information provided by health-care professionals significantly more than those who did not (Table 2). Trust in the Ministry of Health was slightly higher in the adherence group, but the difference between the groups was not significant. The respondents who adhered to COVID-19 protection behaviors had significantly higher mean scores on the conspiracy mentality questionnaire than those who did not.

### 4.5. Predictors of Adherence to COVID-19 Protection Behaviors

The forward likelihood ratio method of logistic regression was implemented to explore the predictors of COVID-19 protection behaviors. The overall model explained 12% of the variance in protective behaviors.

Female gender, being older than 30 years of age, higher level of education, the levels of perceived susceptibility, the perceived proximity, self-efficacy in preventing infection, having the correct knowledge, and acquiring information more than once per day about COVID-19 pandemics were positively related to the COVID-19 protection behaviors.

While the willingness to vaccinate against COVID-19 increased adherence to COVID-19 protection behaviors, engagement with protection behaviors decreased as the conspiracy mentality questionnaire score increased.

High levels of pandemic-related stress and worries about being unable to visit relatives who depend on the respondents were positive predictors of adherence to protective behavior. Other negative emotions, such as fear or worries about mental and physical health issues or financial issues, were not significantly related to adherence to the protective behavior.

It was also determined that trust in hospitals and trust in information provided by health-care professionals were positively related to more adherence to protection behaviors. Unexpectedly, trust in the Ministry of Health showed a small but negative effect on protective behavior. Table 3 shows the predictors of COVID-19 protection behaviors.

The Hosmer–Lemeshow goodness-of-fit test was significant (*p* = 0.739), indicating that the model is correctly specified; additionally, the −2 Log likelihood = 3078.565 and the Nagelkerke R squared = 0.122. Model X^2^ = 293.255 (df = 16), *p* = 0.000).

## 5. Discussion

At the beginning of this study, there was no widespread effective treatment or vaccine. On 14 January 2021, COVID-19 vaccination began in Turkey, and 81.6% of the population was fully vaccinated against COVID-19 in a year, according to the Ministry of Health [27]. Although this rate appears to be satisfactory, it is necessary to adhere to protective behaviors because the vaccines are protective for 6–9 months, and the vaccine response to every new COVID-19 variant will be unknown. As a result, protective behavior remains critical during this period when the numbers of COVID-19 cases and deaths are continuously increasing. In the current study, the potential variables that could influence protection behaviors were investigated.

### 5.1. Sociodemographic Characteristics

There are psychological, social, and behavioral differences in addition to hormones, genes, and immune responses between the sexes that affect COVID-19 progression [28]. In this study, women engaged in more COVID-19 protection behaviors than men, just as they did at the start of the pandemic [29,30]; additionally, younger age groups were less likely to adopt COVID-19 protection behaviors, which was consistent with previous studies [17,31]. The clinical symptoms of COVID-19 manifest less frequently, and background diseases are less common in the younger population than in the older population [32]. As a result, younger age groups may have adopted fewer protective behaviors; thus, a younger demographic structure and lower adherence to protective behavior in the 18–30 age group may pose a critical risk for the spread of COVID-19 in Turkey.

In this study, a modest relationship was found between protective behaviors and education. Previous studies have shown that education is less effective in adopting prevention behavior, but the physical environment and financial resources provided by education level may have contributed to protection behavior [17]; even so, because the majority of our sample consisted of highly educated people, this result should be interpreted with caution.

Vaccinating against COVID-19 appears to cause immunity, but the duration of the immunity or effectiveness of the vaccines for each new COVID-19 variant is unknown. Due to an increase in vaccine refusal and a lack of widely used new treatment options, the best way to control the spread of infections is to adapt COVID-19 protection behaviors. Owing to the apparent predictive effect of gender, age, and education level on protective behaviors, specific plans should be created to strengthen COVID-19 preventive behaviors.

### 5.2. The Perceptions about COVID-19 and Self-Efficacy

Consistent with previous studies, perceived susceptibility to COVID-19 had a positive effect on COVID-19 protection behaviors [33].

In various countries, the level of knowledge is important in adapting to protective behaviors [34]. It was also determined that the frequency of obtaining information and having correct knowledge about COVID-19 were positive predictors of protective behavior in this study; furthermore, self-efficacy was conceptualized in this study as the perceived capability of managing protective behaviors [35,36]. According to current findings, perceived self-efficacy has a positive effect on protective behavior.

### 5.3. Emotional Factors Related to Pandemic

The worry associated with the pandemic could motivate people to engage in protective behaviors [8,37]. The concern for the pandemic should be evaluated from multiple perspectives because the COVID-19 pandemic is a crisis with health, social, and economic implications. In this study, people who engage in protection behavior were more worried about their own financial situations and the country’s economy, but mostly their own health and the health of their immediate surroundings, and being unable to support loved ones when they were in need.; additionally, the latter was an important positive predictor of protection behavior. According to studies, high collectivism indexes at both the personal and national levels positively correlate with infectious disease prevalence [38] and protective behavior [39]; furthermore, protective behavior should be understood within the context of the cultural framework [40]. Turkey is also a collectivist country, which appears to mean that “we” is important rather than “I” [41]. At the individual level, the cultural framework may influence ‘concerning about other people’s health conditions’ and ‘inability to help them when they are in need’. The current findings may guide future studies on the interaction of health behavior with affective and cultural factors.

### 5.4. Trust in Institutions and Beliefs in Conspiracy Theories

Adherence to protective behavior was associated with a higher level of trust in health-care professionals (hospitals and medical professional organizations) and the information they provide. Although the Ministry of Health has demonstrated effective and timely responses since the outbreak began, the variability in data-sharing policy may have influenced public-risk perception and adherence to protective measures [8,37]; however, this subject warrants further exploration.

Since the pandemic is a process dominated by uncertainty and panic, which may increase distrust in institutions [42], it is crucial that official institutions, such as the Ministry of Health, build trust and commitment in the later stages of the pandemic. While restricted access to reliable information increases beliefs in conspiracy theories [43], belief in conspiracy theory reduces trust in institutions and adherence to protective behavior [44]. Consistent with this study, it was determined that belief in conspiracy theories reduces COVID-19 protection behaviors; thus, it is the responsibility of the government and the health authorities to provide access to accurate information; furthermore, the information should be communicated by medical professionals and scientists as a result of these findings.

## 6. Conclusions

Gender, age group, education level, correct knowledge level, and daily frequency of obtaining information were significant predictors of COVID-19 protection behaviors. Transparent, reliable, and culture-specific information sharing by health-care professionals regarding the COVID-19 pandemic and protection behavior is important to increase adherence to COVID-19 protection behaviors; furthermore, health authorities and nongovernmental organizations can help to promote protective behaviors by including programs, especially for young, male, and low-educated individuals, in their public-awareness training.

### Limitations and Strengths

Due to limited internet access, elderly, uneducated, rural, and semiurban citizens were underrepresented in this study. This may make it difficult to determine the representability of the results in the general population; however, these conditions were common for many other studies conducted during the pandemic. Additionally, adherence to protective behaviors was assessed using dichotomous response options in this study. As a result, we were unable to address the influence of the frequency of protective activity. The study’s strengths include the large number of participants and the assessment of public and individual views that may influence protective behaviors. Finally, the findings can help with understanding the aspects of behavior, measuring change strategies, and spreading effective interventions.

## Figures and Tables

**Table 1 vaccines-10-02027-t001:** Adherence to COVID-19 protection behaviors according to sociodemographic variables, knowledge, and exposure to COVID-19 virus.

Adherence to COVID-19 Protection Behaviors	Adherence Group (*n* = 3588)		Nonadherence Group (*n* = 579)		χ^2^	df	*p*
Age (ranges)	N	%	N	%			
18–24 years	1008	81.9	223	18.1	37.09	4	<0.001
25–30 years	475	84.4	88	15.6			
31–39 years	701	87.8	97	12.2			
40–50 years	815	89.2	99	10.8			
51 years and above	632	89.8	72	10.2			
Gender					14.18	1	<0.001
Female	2225	87.9	307	12.1			
Male	1406	83.8	273	16.2			
Education					22.52	1	<0.001
High school or lower	613	80.9	145	19.1			
University graduates and above	3018	87.4	434	12.6			
Family characteristics							
Family size ≥ 5	649	82.7	136	17.3	10.37	1	0.001
Family size ≤ 4	2982	87.1	443	12.9			
Children < 18 years in the household							
Yes	1366	87.5	195	12.5	3.32	1	0.068
No	2265	85.5	384	14.5			
Elderly > 65 years in the household							
Yes	524	87.8	73	12.2	1.44	1	0.229
No	3086	85.9	505	14.1			
Exposure to COVID-19							
Infected participants	125	88.7	16	11.3	0.71	1	0.399
Not infected participants	3506	86.2	563	13.8			
Presence of COVID-19 cases in immediate social environment							
Yes	1776	85.4	304	14.6	2.57	1	0.108
No	1855	87.1	275	12.9			
Presence of chronic disease	513	88.4	67	11.6	2.74	1	0.097
Absence of chronic disease	3118	85.9	512	14.1			
Knowledge on COVID-19 (treatment options and protective behavior)							
Correct knowledge	2982	89.0	367	11.0	107.89	1	<0.001
Lack of knowledge	649	75.4	212	24.6			
The frequency of acquiring information about COVID-19					27.58	2	<0.001
None or once a day	63	67.7	30	32.3			
2–4 times daily	2684	86.6	417	13.4			
5–10 times daily	884	87.0	132	13.0			

**Table 2 vaccines-10-02027-t002:** Adherence to COVID-19 protection behaviors according to knowledge, COVID-19-related emotions, and perceptions.

Adherence to COVID-19 Protection Behaviors	Adherence (*n* = 3588)		Nonadherence (*n* = 579)		t *	*p*
	Mean	Sd	Mean	Sd		
Self-efficacy	5.06	0.82	4.81	0.94	−6.808	<0.001
Cognitive Risk perception						
Susceptibility	4.71	1.74	4.29	1.73	−5.452	<0.001
Severity	4.04	1.64	3.73	1.67	−4.031	0.001
Emotional Risk Perception						
Proximity to COVİD-19 **	4.55	1.61	4.86	1.59	3.245	0.004
The spread of the pandemic (fast/slow)	6.06	1.30	5.69	1.50	−6.066	<0.001
Media hyped	3.04	1.77	3.34	1.91	3.747	<0.001
Fear-inducing	4.88	1.60	4.60	1.75	−3.822	<0.001
Worry	5.39	1.52	5.06	1.66	−4.679	<0.001
Helplessness	3.95	1.91	3.87	1.98	−0.906	0.334
Stressful	5.10	1.59	4.72	1.76	−5.158	<0.001
Worry Domains Questionnaire						
Loss of loved ones	6.17	1.38	5.82	1.66	−5.560	<0.001
Overload of health systems	5.71	1.48	5.26	1.74	−6.689	<0.001
His/her own mental health	4.77	1.87	4.39	2.03	−4.478	<0.001
His/her own physical health	4.89	1.80	4.42	1.97	−5.716	<0.001
Health of loved ones	6.10	1.36	5.78	1.63	−5.241	<0.001
Access to food supplies	4.95	1.92	4.65	1.96	−3.538	0.001
Restrictions on freely travelling	5.23	1.76	5.10	1.83	−1.716	0.086
Unable to go on vacation	3.70	2.13	3.54	2.15	−1.747	0.081
The bankruptcy of small business	5.24	1.76	5.04	1.92	−2.532	0.011
Economic recession	5.79	1.54	5.64	1.75	−2.253	0.024
Job loss	4.36	2.34	4.16	2.37	−1.861	0.063
Unable to pay bills	4.35	2.29	4.12	2.32	−2.234	0.026
Unable to visit people/relatives who depend on them	5.28	1.82	4.83	2.03	−5.445	<0.001
Trust in institutions						
Media	2.68	1.70	2.46	1.66	−3.227	0.001
Medical professional organizations	4.42	2.13	3.99	2.21	−4.478	<0.001
Hospitals	4.72	1.97	4.17	2.02	−5.979	<0.001
Information provided by health-care professionals	5.49	1.57	5.01	1.83	−6.671	<0.001
The Ministry of Health	4.04	2.23	3.87	2.17	−1.731	0.078
Conspiracy Mentality Questionnaire	3.97	1.96	4.19	2.04	2.506	0.012
COVID-19 vaccination willingness	4.88	2.18	4.46	2.35	−4.297	<0.001

* Independent samples *t* test, equality of variances was checked with Levene’s test, df = 4208 ** Lower scores indicate high levels of perceived proximity.

**Table 3 vaccines-10-02027-t003:** The predictors of COVID-19 protection behaviors.

	B	S.E.	Wald	df	Sig.	Exp(B)	95% CI for EXP(B)	
							Lower	Upper
Sociodemographic variables								
Gender	0.337	0.096	12.334	1	<0.001	1.401	1.161	1.691
Education	0.270	0.115	5.497	1	0.019	1.310	1.045	1.641
Being over 30 years of age (reference)			26.442	2	<0.001			
Being 18–24 years of age	−0.525	0.106	24.666	1	<0.001	0.591	0.481	0.728
Being 25–30 years of age	−0.382	0.140	7.461	1	0.006	0.682	0.519	0.898
Correct knowledge about COVID-19	0.866	0.101	72.968	1	<0.001	2.377	1.949	2.899
Acquiring information (less than twice daily) *	−0.636	0.250	6.477	1	0.011	1.889	1.157	3.083
Cognitive risk perceptions								
Self-efficacy	0.178	0.058	9.508	1	0.002	1.195	1.067	1.339
Susceptibility	0.084	0.028	9.162	1	0.002	1.087	1.030	1.148
Perceptions about the pandemics								
Proximity	−0.143	0.033	18.928	1	<0.001	1.154	1.082	1.231
Stressful	0.096	0.029	10.943	1	0.001	1.101	1.040	1.165
Unable to visit people/relatives when needed	0.065	0.025	6.853	1	0.009	1.067	1.016	1.120
Trust in institutions								
Trust in hospitals	0.120	0.030	16.186	1	<0.001	1.128	1.064	1.196
Trust in the Ministry of Health	−0.062	0.027	5.206	1	0.023	0.940	0.891	0.991
Trust in information provided by health-care professionals	0.074	0.029	6.343	1	0.012	1.077	1.017	1.141
Conspiracy Mentality Questionnaire	−0.080	0.025	10.119	1	0.001	0.923	0.878	0.970
COVID-19 vaccination willingness	0.045	0.022	4.249	1	0.039	1.046	1.002	1.092
Constant	−2.684	0.436	37.946	1	<0.001	0.068		

*: Getting less information coded as 1 and getting more information coded as 0.

## Data Availability

The study did not report any data. The corresponding author may share the data upon request.

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
