# Peer review of "The Relationship between COVID-19 Protection Behaviors and Pandemic-Related Knowledge, Perceptions, Worry Content, and Public Trust in a Turkish Sample"

_vaccines, 2022, doi:10.3390/vaccines10122027_

Round 1

Reviewer 1 Report

The article proposed by Kuçukkarapinar et al is interesting and very well written.

A few remarks however:

1. Material & Methods reference is missing in several places.

2. The population studied is substantial but we have no indication of its representability in the general population.

3. When working with such large samples, the use of the p-value is not very appropriate because it will be sensitive to minor effects due to the large size of the population studied. Perhaps you could consider using 95% CIs instead.

Author Response

Thank you very much for your notification. Missing references were added to this section.

  1. Material & Methods reference is missing in several places.

Thank you very much for your notification. Missing references were added to this section.

….The items were used to measure respondents' attitudes toward the COVID-19 pandemic. (Kucukkarapinar et al., 2021)

….The detailed study protocol was described in our previous study: (Kucukkarapinar et al., 2021)

…. Age, sex, educational status, family size, presence of chronic disease (i.e., obesity, diabetes, hypertension, chronic obstructive pulmonary disease), being infected with COVID-19, and knowing people infected with COVID-19 in their intimate social environment were all factors to consider. (Kucukkarapinar et al., 2021)

  1. The population studied is substantial, but we have no indication of its representability in the general population.

       Thank you for your valuable comment, which has been highlighted in the limitations and strengths section.

Limitations and Strengths

Due to limited internet access, elderly, uneducated, rural, and semiurban citizens were underrepresented in this study, this may make it difficult the representability of the results in the general population. However, these conditions were common for many other studies conducted during the pandemic.

  1. When working with such large samples, the use of the p-value is not very appropriate because it will be sensitive to minor effects due to the large size of the population studied. Perhaps you could consider using 95% CIs instead.

Thank you for your valuable comment. Table 2 now includes 95% CIs in addition to the p value. The chi square test was used for the sociodemographic data in Table 1, but we did not think it was necessary to include it because the 95% CIs value is not widely used here.

Table 2 Adherence to COVID-19 protection behaviors according to knowledge, COVID-19-related emotions, and perceptions.

Reviewer 2 Report

The study is well conducted, linked to the relevant literature, and results are convincing. My only concern has to do with the changes in respondents over the period in which the survey was conducted:  between July 2020 and 29 January 2021. In this period the situation changed from a moment in which there were no expectation of vaccines to a moment in which an announcement was made (Autumn 2020) and vaccination started in many countries worldwide. These dynamics might have affected the responses of surveyed agents taken at different moments in time. My suggestion is to show if the results are robust to the different timeline of responses, in particular as regards availability/announcements about effective vaccines, as perceptions of agents might have changed over time.

Author Response

Thank you very much for your suggestions and comments. COVID-19 vaccination began in Turkey on January 14, 2021. Priority was given at this time to healthcare professionals and individuals over the age of 65. As we emphasized in the manuscript, the time between the start of vaccination studies and the first vaccination is included in our research data. However, we did not include data on this because when we analyzed the data, we did not find a significant change in perceptions over time.

Reviewer 3 Report

The global COVID-19 pandemic is causing a public health crisis. The vaccination protects people from the illness caused by the SARS-CoV-2 infection. However, due to the continuing emergence of new variants, the vaccine's efficacy is not good enough to prevent SARS-CoV-2 infection and spread. As a result, hygiene practices and maintaining physical distance continue to play an important role in COVID-19 control. By analyzing online survey data (Turkish COVID-19 Snapshot Monitoring) between July 2020 and January 2021, the authors of this study attempted to investigate the effect of knowledge, COVID-19-related perceptions, and public trust on protective behaviors in Turkish people. They found that 13.8% of the 4,210 adult respondents reported nonadherence to protection behavior, whereas 86.2% reported full adherence. Males and young people (18-30 years old) have lower levels of adherence. Perceived stress, altruistic worries, and public trust appear to impact protective activities, along with people's knowledge and cognitive risk perception in responders. 

Author Response

We greatly appreciate your opinions on our study.